# Short- and Long-Term Outcomes of Patients with Postoperative Arrhythmia after Liver Surgery

**DOI:** 10.3390/biomedicines12020271

**Published:** 2024-01-25

**Authors:** Felix Rühlmann, Deborah Engelhardt, Alma Franziska Mackert, Mara Sophie Hedicke, Tobias Tichelbäcker, Andreas Leha, Markus Bernhardt, Michael Ghadimi, Thorsten Perl, Azadeh Azizian, Jochen Gaedcke

**Affiliations:** 1Department of General, Visceral and Paediatric Surgery, University Medical Centre Göttingen, 37075 Göttingen, Germany; felix.ruehlmann@med.uni-goettingen.de (F.R.); e.deborahchristina@stud.uni-goettingen.de (D.E.); alma.mackert@med.uni-goettingen.de (A.F.M.); mara.hedicke@med.uni-goettingen.de (M.S.H.); markus.bernhardt@med.uni-goettingen.de (M.B.); mghadim@uni-goettingen.de (M.G.); thorsten.perl@med.uni-goettingen.de (T.P.); azadeh.azizian@med.uni-goettingen.de (A.A.); 2Clinic III for Internal Medicine, Heart Centre of University Hospital of Cologne, 50937 Cologne, Germany; tobias.tichelbaecker@uk-koeln.de; 3Institute of Medical Statistics, University Medical Centre Göttingen, 37075 Göttingen, Germany; andreas.leha@med.uni-goettingen.de

**Keywords:** liver surgery, postoperative new-onset arrhythmia, perioperative medicine, perioperative risk stratification

## Abstract

Background: New-onset postoperative arrhythmia (PA) has previously been described as a pivotal risk factor for postoperative morbidity and mortality after visceral surgery. However, there is a lack of data concerning liver surgery. The incidence and impact of new-onset postoperative arrhythmia after liver surgery was, therefore, analyzed in a monocentric study. Methods: In total, *n* = 460 patients (221 female, 239 male) who underwent liver surgery between January 2012 and April 2020 without any prior arrhythmia in their medical history were included in this retrospective analysis. Clinical monitoring started with the induction of anesthesia and was terminated with discharge from the intensive care unit (ICU) or intermediate care unit (IMC). Follow-up included documentation of complications during the hospital stay, as well as long-term survival analysis. Results: Postoperative arrhythmia after liver surgery was observed in 25 patients, corresponding to an incidence of 5.4%. The occurrence of arrhythmia was significantly associated with intraoperative complications (*p* < 0.05), liver fibrosis/cirrhosis (*p* < 0.05), bile fistula/bile leakage/bilioma (*p* < 0.05), and organ failure (*p* < 0.01). Survival analysis showed a significantly poorer overall survival of patients who developed postoperative arrhythmia after liver surgery (*p* < 0.001). Conclusions: New-onset postoperative arrhythmia after liver surgery has an incidence of only 5.4% but is significantly associated with higher postoperative morbidity and poorer overall survival.

## 1. Introduction

Postoperative arrhythmia (PA) is an underrecognized complication after visceral surgery. Recently, we analyzed the occurrence of PA after surgical procedures involving the esophagus, stomach and pancreas with a retrospective study design and showed an incidence of 8.3% of the total patients, with a vast difference between the surgical procedures. Moreover, PA was significantly associated with in-house mortality and severe postoperative complications [1]. These findings were in line with those of a subsequent study of PA after surgery on the lower gastrointestinal tract: While the incidence was distinctively lower, the impact remained severe [2].

Among the existing studies concerning PA after visceral surgery [3,4], most have focused on esophageal surgery [5,6,7], presumably because it is performed partly intrathoracically and has a high rate of postoperative morbidity and mortality.

Comparable to esophageal surgery, major liver surgery also results in high rates of postoperative morbidity and mortality. Some general risk factors are already well described, but data concerning PA and general liver surgery have not yet been published. Only the relevance of PA after liver transplantation has been described in some studies [8,9,10].

Liver surgery does not have a thoracic component, but the anatomical proximity and the functional dependence between the heart and liver necessitate an analysis of PA after liver surgery. Additionally, PA has often been associated with postoperative complications (e.g., infection, organ failure, mortality) [1,2] that regularly occur after liver surgery.

In the present study, our aim was to evaluate the incidence and impact of PA after liver surgery and identify associated risk factors.

We present this article in accordance with the STROBE reporting checklist.

## 2. Materials and Methods

### 2.1. Patient Cohort

For this study, we collected data from *n* = 528 patients who underwent liver surgery in the period from January 2012 to April 2020 in the Department of General, Visceral, and Pediatric Surgery, University Medical Centre Göttingen, Germany. Patients with a preexisting cardiac arrhythmia (*n* = 58) or a previously implanted pacemaker (*n* = 16) were excluded from further analysis. We included only patients who went to the ICU/IMC after surgery and were monitored for at least 24 h.

Overall, data from *n* = 460 patients were included in the present study. The data included age, sex, type of surgical procedure, preexisting conditions (e.g., comorbidities), intraoperative complications (e.g., iatrogenic injury or blood transfusion requirement after strong bleeding), main diagnoses, postoperative complications, duration of ICU stay, and in-house mortality, as well as general mortality. Preexisting cardiac illness was classified as present if patients had a myocardial infarction, coronary heart disease, heart failure, or a valvular heart disease in the past. The classification of the extent of liver resection was based on that of Reddy et al. [11]: Operations in which fewer than four liver segments were resected were considered minor resections (vs. four or more segments in major resections). HCC and intrahepatic CCC were partly grouped as “primary liver cancer”. Other tumor entities infiltrating the liver without any origin of liver parenchyma (gallbladder carcinoma, perihilar cholangiocarcinoma, or distal bile duct carcinoma) were grouped as ‘liver-infiltrating malignant tumors’. Regarding the definition of organ failure as a possible postoperative complication, a subdivision into liver, lung, and kidney failure was made. According to the “50–50 Criteria” [12], postoperative liver failure correlates with a prothrombin time below 50% on the fifth postoperative day and, at the same time, a serum bilirubin level of more than 50 μmol/L (equals 2.94 mg/dL). Lung failure was defined as moderate or severe acute respiratory distress syndrome (ARDS) with a Horowitz quotient or oxygenation index of 200 mmHg or less according to the Berlin definition [13]. Finally, acute kidney failure was determined based on the KDIGO guidelines [14]. In order to identify patients with sepsis, first, the qSOFA score was calculated as the sum of the following (fulfilled) clinical criteria [15]:Respiratory frequency: ≥22/min;Systolic blood pressure: <100 mmHg;CNS: reduced vigilance or altered mental status (GCS score);

If at least two of these criteria applied to a patient, septic organ dysfunction was suspected, and the simultaneous presence of infections (documented by a doctor in the patient file) and organ failure were checked as further evidence.

For all patients who had developed PA, we conducted follow-up screening for permanent arrhythmia and thromboembolic events. We contacted the patients, their family doctors, and, if available, their cardiologists. A standardized survey was used (Appendix A shows the translated version of the German survey). Overall survival (OS) was analyzed for all included patients if the data were available. Existing medical records and published obituaries served as proof of death. The minimum survival time (in months) was calculated as the difference between the day of surgery and the day of death. If the death of a patient could not be traced, the last patient visit at the University Medical Center Göttingen (e.g., a presentation in the outpatient clinic or the end of another inpatient stay) was used as the point in time up to which the minimum survival could be calculated. Survival time analyses were subsequently carried out.

### 2.2. Statistical Analysis

All statistical analyses were performed using the statistical programming environment “R” (version 4.1.2.; R Foundation for Statistical Computing, Vienna, Austria). Descriptive data are summarized in tables with absolute and relative frequencies (n/number of cases) for categorical variables. For continuous variables, mean values (including the standard deviation), medians (with interquartile range), maxima, and minima are reported in tables or in the text. The significance level was set to 0.05 for all analyses.

Pairwise comparisons between nominally scaled variables were conducted using the chi-square test or Fisher’s exact test (Fisher–Yates test) if the expected number of events in one field of the four-field table was less than 5. For ordinal variables, two-group comparisons were conducted using the Mann–Whitney U test (Wilcoxon rank sum test). Means for interval-scaled data from two patient groups were compared using two-tailed *t*-tests.

The effects of different diagnosis groups on the occurrence of PA were assessed using one-way ANOVA and Tukey’s post hoc test to compare all possible group combinations.

Multivariable logistic regression models were fit to the data to model the occurrence of PA and in-hospital mortality. The relationship between each independent variable and the occurrence of PA in-hospital mortality was examined in advance (univariable analyses), and variables showing no associations (*p* > 0.05) with in-hospital mortality were excluded from the multivariable logistic regression modeling. The selection of which variables were ultimately relevant predictors was carried out using a step-down procedure in the logistic regression model fitting. This greedily minimized the Akaike information criterion (AIC) by excluding variables (in a stepwise procedure) to reduce the prediction error.

From the final model, the odds ratios are presented graphically with 95% confidence intervals (CIs).

Survival curves were estimated and are presented graphically using the Kaplan–Meier estimator. Using the log-rank test, the survival curves of patients with malignant and benign diseases and those of patients with different types of liver tumors were compared.

## 3. Results

### 3.1. Demographic Data

Overall, *n* = 460 patients who underwent liver surgery were included, among whom *n* = 239 were male (52%) and *n* = 221 were female (48%). The mean (±SD) age of the subjects was 60.9 ± 13.6 years (range: 19–87 years). Surgical procedures included *n* = 163 hemihepatectomies on the right, *n* = 64 hemihepatectomies on the left, *n* = 43 segmentectomies, *n* = 41 right trisegmentectomies, *n* = 46 multivisceral resections, *n* = 47 bisegmentectomies (mainly left lateral and right posterior sectorectomies), *n* = 31 resections of other segment combinations, *n* = 12 atypical liver resections or wedge resections, *n* = 11 “in situ split” liver resections (with subsequent trisegmentectomy or hemihepatectomy), and *n* = 2 left trisegmentectomies. A total of 387 patients (84%) were diagnosed with malignant neoplasia. Specific diagnoses, comorbidities, perioperative complications, and other individual parameters are listed in Table 1.

### 3.2. Incidence, Time Point, and Type of PA

In total, 25 of 460 patients developed PA (5.4%) after liver surgery, with a wide range between the different surgical procedures (Table 1). The difference in incidence between the surgical procedures, however, remained statistically nonsignificant. Table 1 shows all surgical procedures and the occurrence of PA in the respective groups.

The detected types of PA were mostly tachycardic arterial fibrillation (*n* = 14, 56%), asystole (*n* = 5, 20%), and ventricular tachycardia (*n* = 3, 12%). Only one patient developed bradycardic arterial fibrillation (4%). In *n* = 3 cases, PA led to cardiopulmonary resuscitation (12%). The mean time interval between liver surgery and the occurrence of PA was 186.1 h (median (Mdn) = 132; interquartile range = 40–312; Min = 0; Max = 672).

### 3.3. PA and Associated Factors

The incidence of PA differed significantly based on the diagnosis (*F*(3456) = 7.72, *p* < 0.001); Tukey post hoc analysis revealed a significantly higher incidence of PA in patients with primary liver cancer (HCC/intrahepatic CCC) than in patients with liver metastases. Additionally, comparing primary liver cancer patients to those with benign lesions, the latter showed a significantly lower incidence of PA (*p* < 0.05).

Patients with liver-infiltrating malignant tumors showed a significantly higher rate of PA than those with benign lesions and patients with liver metastases (Table 2).

To evaluate the impact of the extent of liver tissue resection per se, we pooled all cases into major and minor liver resections. Here, no significant differences in the occurrence of PA were observed between the two groups.

Further univariate tests were carried out to examine associations between the different variables listed in Table 1 and the occurrence of postoperative cardiac arrhythmia.

All variables that were significantly associated with the occurrence of PA were included in a logistic regression model, and those with the greatest explanatory power were extracted using logistic regression analysis. Here, intraoperative complications, liver fibrosis/cirrhosis, organ failure, ascites (defined as 500 > mL/24 h drainage after 3 postoperative days), and bile fistula were identified as significantly associated factors. Figure 1 shows the odds ratios with 95% confidence intervals of the variables in the stepwise logistic regression model for the postoperative development of arrhythmia.

In the further course after liver surgery, the occurrence of PA was also significantly associated with a longer stay in the ICU/IMC (Mdn: 19 days versus 5 days; *p* < 0.01). The mean ICU/IMC stay for all patients was 9.8 days (Mdn = 5 days).

### 3.4. PA and In-House Mortality

In total, *n* = 20 (4.3%) patients died after liver surgery during their inpatient stay. To test the association between clinical variables and in-house mortality, univariate tests were performed. Significant associations were detected between in-house mortality and new-onset PA, organ failure, sepsis, infection, electrolyte disorder, ascites, revision surgery, bile fistula/bile leakage/bilioma, CHA_2_DS_2_-VASc score ≥ 4, current (postoperative) thrombosis, anastomosis/stump insufficiency, age, extent of liver resection, and intraoperative complications (see Table 3).

A forest plot (Figure 2) is used to present the results of the stepwise logistic regression model for in-house mortality with their odds ratios, 95% confidence intervals, and *p*-values.

### 3.5. Long-Term Overall Survival (OS)

Altogether, the follow-up data of 387 of 460 patients (84.1% of all patients) could be obtained, and *n* = 73 were lost to follow-up, as there was no access to any information after their discharge. The median survival time of all patients was 54 months (IQR = 43–69.1), and the median follow-up time was 41.3 months (IQR = 36.8–46.5).

The survival of *n* = 329 patients with liver surgery for malignant disease was compared with the survival of *n* = 58 patients with a benign underlying disease. In the postoperative period up to a maximum of 118.6 months after liver surgery, a total of *n* = 8 (13.8%) of the patients with a benign underlying disease died, and almost half of the patients with a malignant disease died; *n* = 149 (45.3%). In a direct comparison, the survival of patients with a benign underlying disease (Mdn = NA, 95% CI [85.3; NA]) was longer than the survival of patients with a malignant disease (Mdn = 45.3, 95% CI [39.3; 59.2]). A log-rank test was performed and showed significant differences between the survival distributions of both patient groups; *χ*^2^ (1) = 14.1, *p* < 0.001.

Moreover, a comparison was made between patients diagnosed with hepatocellular carcinoma (HCC), cholangiocellular carcinoma (CCC), and colorectal liver metastases (CRLMs). Among the 53 patients with HCC, 43 (81.1%) could be tracked, in addition to 60 of 72 patients (83.3%) with CCC and 169 of 200 patients (84.5%) with CRLM. The median overall survival of patients with HCC was 69.1 months (95% CI [31.4; NA]), compared to 47 months (95% CI [39.5; 60.3]) for patients with CRLM. Thus, the two diagnosis groups, CRLM and HCC, both survived significantly longer than patients with CCC (Mdn = 33.3, 95% CI [14.9; 61.2]; *p* < 0.05). All other comparisons between these diagnosis groups were not significant.

Patients without PA survived significantly longer than those with PA after liver surgery (median 56.8 months vs. 11.3 months, *p* < 0.001). The difference between the survival curves of patients with vs. without PA shown in Figure 3 was significant; *χ*^2^(1) = 26.9, *p* < 0.001.

## 4. Discussion

New onset of cardiac arrhythmia is typically considered a cardiac-surgery-related phenomenon. Little attention has been given to its incidence and impact in noncardiac surgery, although its relevance is known [16]. This may be due to a poor understanding of the pathophysiological process. Chung et al. postulated an inflammatory response triggering disorganized electrical activity within atrial myocytes [17]. These data are in line with our previous data showing a significant association of PA after surgery on the upper and lower gastrointestinal tract with postoperative complications (e.g., infections, sepsis, and organic failure). On the other hand, reduced organ perfusion based on PA is under discussion, leading to the debate of whether PA precedes complications or if it is just a symptom.

Subramani et al. recently published a systematic review and meta-regression analysis of observational studies confirming postoperative atrial fibrillation as a risk factor for postoperative cardiac complications, stroke, and higher mortality. Advanced age, male sex, preoperative hypertension, diabetes mellitus, and cardiac disease were identified as important risk factors for perioperative atrial fibrillation. Faced with an aging and comorbid surgical population, the need for risk stratification and close monitoring increases [18].

In visceral surgery, the type of surgery appears to have an impact on the incidence of PA and may be associated with the intensity of trauma. Interestingly, in liver surgery, the incidence of PA is not associated with the extent of parenchymal removal. Data on liver surgery and PA are rare. Such data are typically collected in liver transplantation, showing arrhythmias as the main cardiovascular disease complication within 30 days after liver transplantation [19,20]. A poorer postoperative outcome in pretransplant arterial fibrillation patients was shown by Dangl et al., who analyzed 45,357 patients who underwent orthotopic liver transplantation [21]. The prevalence of preexisting atrial fibrillation and/or incidence of AF following liver transplantation was analyzed by Chokesuwattanaskul et al. in a meta-analysis enrolling 38,586 liver transplant patients, also indicating a poorer clinical outcome [22]. Rivas et al. published data on 857 patients, which revealed a new onset of postoperative atrial fibrillation (POAF) in 10.4% of patients. While POAF was not associated with in-hospital mortality, the one-year mortality was increased [23].

Posttransplant reperfusion causing hemodynamic stress can result in hemodynamic instability along with arrhythmia [24,25,26]. Reperfusion effects, however, were not attributed to the analyzed patients of the present study, as the Pringle maneuver was not applied. Nevertheless, these findings may confirm the significant association in the present study between PA and intraoperative complications/transfusions of erythrocyte concentrates, as this also represents a process of reperfusion.

The association of PA and preoperative conditions such as older age and preexisting multiple cardiovascular diseases confirms the abovementioned data from Subramani et al. It is of further importance that the diagnosis of primary liver cancer is associated with an increased risk of PA compared to that in cases of liver metastases from colorectal cancer. It can be hypothesized that underlying circumstances leading to primary cancer in the liver are also a trigger for the new onset of PA.

A central finding in the present study is that PA is significantly associated with a poorer OS. This trend was shown in liver transplant patients as outlined above. In patients undergoing surgery for solid tumors, this finding has not yet been published. It is very notable that the occurrence of PA is independently associated with a poorer prognosis. Here, we used a Cox regression model, and both variables, PA and dignity, were significantly associated with poorer survival—each of them as independent variables. The association of PA and poorer survival is of interest, as, e.g., atrial fibrillation itself is not associated with poorer prognosis in the general population. It could be speculated that the occurrence of PA after surgery for liver tumors reveals a previously undetected cardiac disease. On the other hand, fluid management during and after the surgical procedure may have an effect on cardiac function and integrity. As intraoperative low-volume therapy and postoperative high-volume therapy are cornerstones of liver surgery, these volume loads may also trigger a previously undetected heart frailty.

Due to the retrospective nature of this study, no causalities may be discovered here. However, whether as a cause or just as an indicator, PA after liver surgery is significantly associated with postoperative complications and poorer overall survival. Therefore, its occurrence is an important clinical finding, which should be recognized and raise awareness for a possibly critical situation of the patient.

Whether a poorer prognosis of patients with PA is due to the development of undetected heart disease or the result of a rather complicated postoperative course remains unknown.

## 5. Conclusions

To the best of our knowledge, we are the first to report the association between liver surgery and postoperative arrhythmia. Despite its low incidence of 5.4% after liver surgery (compared to other types of visceral surgery), PA is associated with a set of postoperative complications, longer ICU/IMC stay, and poorer overall survival. Due to the retrospective character of this analysis, no conclusion regarding causalities can be stated. Further analyses, especially those regarding causalities, remain subjects for a prospectively designed study.

## Figures and Tables

**Figure 1 biomedicines-12-00271-f001:**
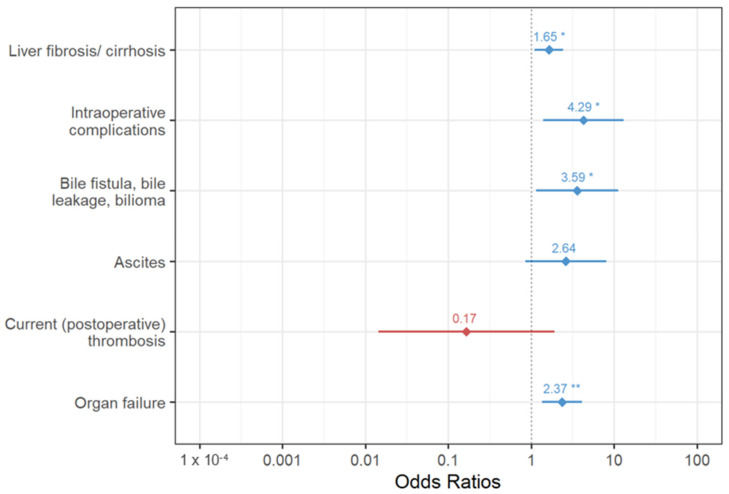
Odds ratios with 95% confidence intervals of variables in the final logistic regression model for the postoperative development of arrhythmia (asterisks indicate the level of significance: *p* < 0.05 *, *p* < 0.01 **).

**Figure 2 biomedicines-12-00271-f002:**
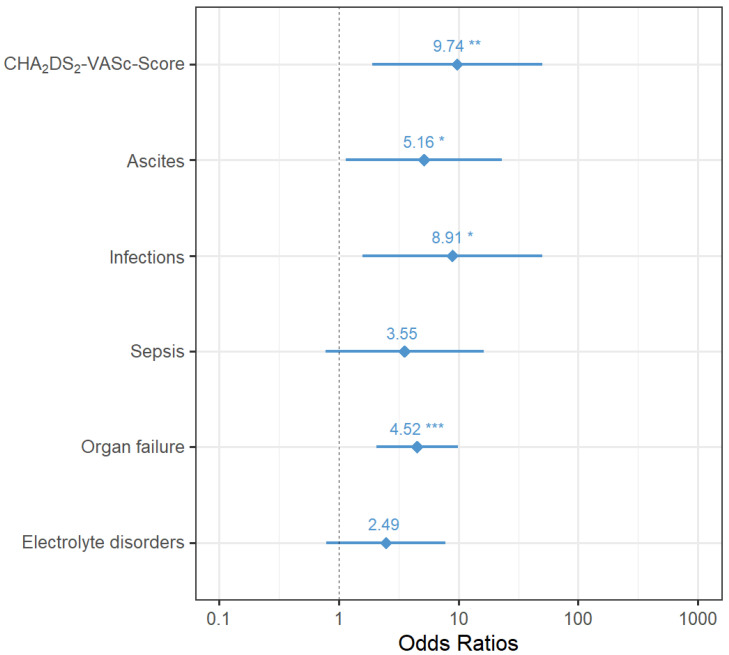
Odds ratios with 95% confidence intervals of variables in the final logistic regression model for postoperative in-house mortality (asterisks indicate the level of significance: *p* < 0.05 *, *p* < 0.01 **, *p* < 0.001 ***).

**Figure 3 biomedicines-12-00271-f003:**
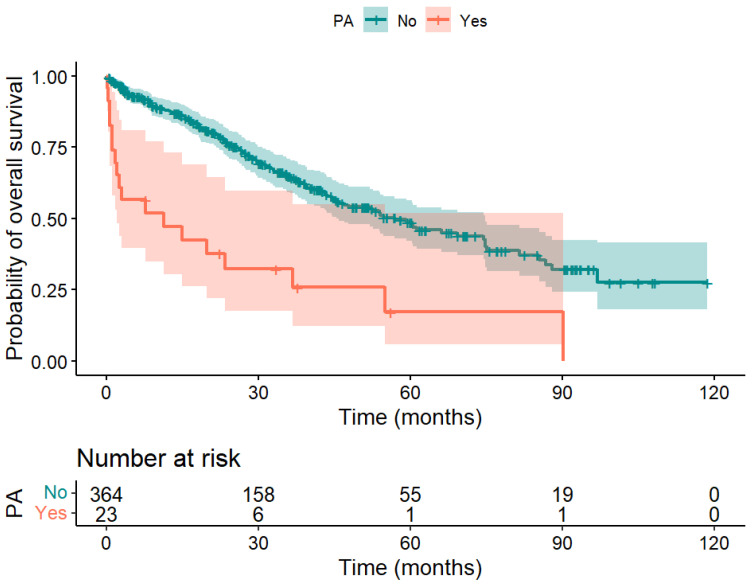
Kaplan–Meier curve of patients with PA (red) and without PA (blue) after liver surgery. A significant difference in overall survival was obtained; *χ*^2^(1) = 26.9, *p* < 0.001.

**Table 1 biomedicines-12-00271-t001:** Descriptive values of the recorded parameters in all included patients undergoing liver surgery and in patients with PA.

Parameter	Level	All Patients *n* (%)	Patients without PA *n* (%)	Patients with PA *n* (%)	*p*
*n*		460 (100)	435 (94.6)	25 (5.4)	
Sex	Male Female	239 (52.0) 221 (48.0)	225 (94.1) 210 (95.0)	14 (5.9) 11 (5.0)	0.838
Age	Mean ± sd	60.9 ± 13.6	60.7 ± 13.8	65.3 ± 9.9	0.033
	Median (min; max)	63.6 (19.2; 87.1)	63.6 (19.2; 87.1)	64.0 (35.9; 80.9)	
Surgery					n.s.
	(Extended) right hemihepatectomy	163 (35.4)	152 (93.3)	11 (6.7)	
	(Extended) left hemihepatectomy	64 (13.9)	62 (96.9)	2 (3.1)	
	Segmentectomy	43 (9.3)	42 (97.7)	1 (2.3)	
	Right trisegmentectomy	41 (8.9)	38 (92.7)	3 (7.3)	
	Multivisceral resection	41 (8.9)	39 (95.1)	2 (4.9)	
	Cytoreductive surgery with HIPEC	5 (1.1)	4 (80.0)	1 (20.0)	
	Bisegmentectomy (e.g., left lateral and right posterior sectionectomy)	47 (10.2)	45 (95.7)	2 (4.3)	
	Resection of other segment combinations	31 (6.7)	29 (93.5)	2 (6.5)	
	Atypical liver resections, wedge resections	12 (2.6)	12 (100.0)	0 (0.0)	
	ALLPS	11 (2.4)	11 (100.0)	0 (0.0)	
	Left trisegmentectomy	2 (0.4)	1 (50.0)	1 (50.0)	
Major resection	No Yes	193 (42.0) 267 (58.0)	185 (95.9) 250 (93.6)	8 (4.1) 17 (6.4)	0.407
Diagnosis	Benign disease	73 (16.0)	71 (97.3)	2 (2.7)	0.400
	Malignant disease	387 (84.0)	364 (94.1)	23 (5.9)	
	- CRLM	200 (43.5)	196 (98.0)	4 (2.0)	
	- Other liver metastases	29 (6.3)	29 (100.0)	0 (0.0)	
	- Nonvital metastases	5 (1.1)	5 (100.0)	0 (0.0)	
	- HCC	53 (11.5)	47 (88.7)	6 (11.3)	
	- Intrahepatic CCC	46 (10.0)	41 (89.1)	5 (10.9)	
	- Extrahepatic CCC	26 (5.7)	22 (84.6)	4 (15.4)	
	- Combined HCC/CCC	4 (1.0)	3 (75.0)	1 (25.0)	
	- Other infiltrating tumors	24 (5.2)	21 (87.5)	3 (12.5)	
Dyslipidemia	No Yes	408 (88.7) 52 (11.3)	386 (94.6) 49 (94.2)	22 (5.4) 3 (5.8)	0.754
Hypertension	No Yes	223 (48.5) 237 (51.5)	214 (96.0) 221 (93.2)	9 (4.0) 16 (6.8)	0.281
Diabetes mellitus	No Yes (Type I/II)	382 (83.0) 78 (17.0)	364 (95.3) 71 (91.0)	18 (4.7) 7 (9.0)	0.165
Venous thrombosis	No Yes (current) Yes (in past)	405 (88.0) 23 (5.0) 32 (7.0)	384 (94.8) 19 (82.6) 32 (100.0)	21 (5.2) 4 (17.4) 0 (0.0)	0.030
Embolic event	No Yes (apoplex/ transient ischemic attack in past)	441 (95.9) 19 (4.1)	417 (94.6) 18 (94.7)	24 (5.4) 1 (5.3)	0.662
Preexisting cardiac illness	No Yes	397 (86.3) 63 (13.7)	377 (95.0) 58 (92.1)	20 (5.0) 5 (7.9)	0.37
Multiple (≥2) cardiovascular diseases	No Yes	304 (66.1) 156 (33.9)	293 (96.4) 142 (91.0)	11 (3.6) 14 (9.0)	0.029
Kidney diseases	No Yes	420 (91.3) 40 (8.7)	398 (94.8) 37 (92.5)	22 (5.2) 3 (7.5)	0.469
Intraoperative complications	No Yes	363 (78.9) 97 (21.1)	349 (96.1) 86 (88.7)	14 (3.9) 11 (11.3)	0.009
Anastomosis insufficiency	No Yes	447 (97.2) 13 (2.8)	422 (94.4) 13 (100.0)	25 (5.6) 0 (0.0)	1
Wound healing deficit	No Yes	391 (85.0) 69 (15.0)	373 (95.4) 62 (89.9)	18 (4.6) 7 (10.1)	0.079
Bile fistula, bile leakage, bilioma	No Yes	369 (80.2) 91 (19.8)	354 (95.9) 81 (89.0)	15 (4.1) 10 (11.0)	0.017
Liver parenchyma	Normal	163 (35.4)	158 (96.9)	5 (3.1)	0.004
	Steatosis	145 (31.5)	141 (97.2)	4 (2.8)	
	Fibrosis	43 (9.3)	38 (88.4)	5 (11.6)	
	Cirrhosis	26 (5.7)	21 (80.8)	5 (19.2)	
	Not classified	83 (18.0)	77 (92.8)	6 (7.2)	
Ascites (500 > mL/24 h drainage after the third postoperative day)	No Yes	355 (77.2) 105 (22.8)	344 (96.9) 91 (86.7)	11 (3.1) 14 (13.3)	<0.001
Pleural effusion	No Yes	361 (78.5) 99 (21.5)	350 (97.0) 85 (85.9)	11 (3.0) 14 (14.1)	<0.001
Revision surgery	No Yes	415 (90.2) 45 (9.8)	398 (95.9) 37 (82.2)	17 (4.1) 8 (17.8)	0.002
Organic failure	No Yes	317 (68.9) 143 (31.1)	309 (97.5) 126 (88.1)	8 (2.5) 17 (11.9)	<0.001
Electrolyte disorders	No Yes	254 (55.2) 206 (44.8)	248 (97.6) 187 (90.8)	6 (2.4) 19 (9.2)	<0.001
Postoperative delirium	No Yes	411 (89.4) 49 (10.6)	391 (95.1) 44 (89.8)	20 (4.9) 5 (10.2)	0.170
Infections	No Yes	337 (73.3) 123 (26.7)	324 (96.1) 111 (90.2)	13 (3.9) 12 (9.8)	0.025
Sepsis	No Yes	433 (94.1) 27 (5.9)	417 (96.3) 18 (66.7)	16 (3.7) 9 (33.3)	<0.001
CHA_2_DS_2_-VASc-Score ≥4	No Yes	376 (81.7) 84 (18.3)	358 (95.2) 77 (91.7)	18 (4.8) 7 (8.3)	0.190
Betablocker preoperatively	No Yes	325 (70.7) 135 (29.3)	311 (95.7) 124 (91.9)	14 (4.3) 11 (8.1)	0.160

**Table 2 biomedicines-12-00271-t002:** List of all significant differences between the diagnosis groups regarding the incidence of PA in Tukey post hoc analysis with the median, 95% confidence interval, and *p*-values (asterisks indicate the level of significance: *p* < 0.05 *, *p* < 0.01 **, *p* < 0.001 ***). Liver-infiltrating malignant tumors (without origin of liver parenchyma): tumors such as gallbladder carcinoma, perihilar cholangiocarcinoma or distal bile duct carcinoma.

Diagnosis Groups	PA Incidence		*p*
	Median Difference (in %)	95% CI (in %)	
Patients with primary liver cancer vs. patients with liver metastases	−9.9 (11.6 vs. 1.7)	−3.2–(−16.7)	<0.001 ***
Patients with primary liver cancer vs. patients with benign lesions	−8.9 (11.6 vs. 2.7)	−0.1–(−17.7)	0.04 *
Patients with liver-infiltrating malignant tumors vs. patients with benign lesions	−11.3 (14.0 vs. 2.7)	−0.7–(−21.8)	0.03 *
Patients with liver-infiltrating malignant tumors vs. patients with liver metastases	−12.3 (14.0 vs. 1.7)	−3.4–(−21.2)	0.002 **

**Table 3 biomedicines-12-00271-t003:** Descriptive values of the recorded parameters of all included patients undergoing liver surgery and those of patients who died during their hospital stay.

Parameter	Level	All Patients *n* (%)	Patients Who Did Not Die during Their Stay *n* (%)	Patients Who Died during Their Stay *n* (%)	*p*
*n*		460 (100)	440 (95.7)	20 (4.3)	
Sex	Male Female	239 (52.0) 221 (48.0)	227 (95.0) 213 (96.0)	12 (5.0) 8 (4.0)	0.503
Age	Mean ± sd	60.9 ± 13.6	60.7 ± 13.7	66.8 ± 10.1	0.015
	Median (min; max)	63.6 (19.2; 87.1)	63.5 (52.4; 70.6)	67.8 (59.1; 73.4)	
Surgery					
	(Extended) right hemihepatectomy	163 (35.4)	152 (93.3)	11 (6.7)	
	(Extended) left hemihepatectomy	64 (13.9)	61 (95.3)	3 (4.7)	
	Segmentectomy	43 (9.3)	43 (100.0)	0 (0.0)	
	Right trisegmentectomy	41 (8.9)	40 (97.6)	1 (2.4)	
	Multivisceral resection	41 (8.9)	38 (92.7)	3 (7.3)	
	Cytoreductive surgery with HIPEC	5 (1.1)	5 (100.0)	0 (0.0)	
	Bisegmentectomy (e.g., left lateral and right posterior sectionectomy)	47 (10.2)	47 (100.0)	0 (0.0)	
	Resection of other segment combinations	31 (6.7)	31 (100.0)	0 (0.0)	
	Atypical liver resections, wedge resections	12 (2.6)	12 (100.0)	0 (0.0)	
	In situ split with subsequent trisegmentectomy or (extended) hemi-hepatectomy on the right	11 (2.4)	10 (90.9)	1 (9.1)	
	Left trisegmentectomy	2 (0.4)	1 (50.0)	1 (50.0)	
Major resection	No Yes	193 (42.0) 267 (58.0)	190 (98.4) 250 (93.6)	3 (1.6) 17 (6.4)	0.018
Diagnosis	Benign disease	73 (16.0)	71 (97.3)	2 (2.7)	0.754
	Malignant disease	387 (84.0)	364 (95.3)	18 (4.7)	
	- CRLM	200 (43.5)	199 (99.5)	1 (0.5)	
	- Other liver metastases	29 (6.3)	28 (96.6)	1 (3.4)	
	- Nonvital metastases	5 (1.1)	5 (100.0)	0 (0.0)	
	- HCC	53 (11.5)	51 (96.2)	2 (3.8)	
	- Intrahepatic CCC	46 (10.0)	43 (93.5)	3 (6.5)	
	- Extrahepatic CCC	26 (5.7)	20 (76.9)	6 (23.1)	
	- Combined HCC/CCC	4 (1.0)	4 (100.0)	0 (0.0)	
	- Other infiltrating tumors	24 (5.2)	19 (79.2)	5 (20.8)	
Dyslipidemia	No Yes	408 (88.7) 52 (11.3)	389 (95.3) 51 (98.1)	19 (4.7) 1 (1.9)	0.714
Hypertension	No Yes	223 (48.5) 237 (51.5)	216 (96.9) 224 (94.5)	7 (3.1) 13 (5.5)	0.315
Diabetes mellitus	No Yes (type I/II)	382 (83.0) 78 (17.0)	367 (96.1) 73 (93.6)	15 (3.9) 5 (6.4)	0.357
Venous thrombosis	No Yes (current) Yes (in past)	405 (88.0) 23 (5.0) 32 (7.0)	389 (96.0) 19 (82.6) 32 (100.0)	16 (4.0) 4 (17.4) 0 (0.0)	0.014
Embolic event	No Yes (apoplex/ transient ischemic attack in past)	441 (95.9) 19 (4.1)	422 (95.7) 18 (94.7)	19 (4.3) 1 (5.3)	0.578
Preexisting cardiac illness	No Yes	397 (86.3) 63 (13.7)	379 (95.5) 61 (96.8)	18 (4.5) 2 (3.2)	1
Multiple (≥2) cardiovascular diseases	No Yes	304 (66.1) 156 (33.9)	295 (97.0) 145 (92.9)	9 (3.0) 11 (7.1)	0.073
Kidney diseases	No Yes	420 (91.3) 40 (8.7)	402 (95.7) 38 (95.0)	18 (4.3) 2 (5.0)	0.689
Intraoperative complications	No Yes	363 (78.9) 97 (21.1)	352 (97.0) 88 (90.7)	11 (3.0) 9 (9.3)	0.020
Anastomosis insufficiency	No Yes	447 (97.2) 13 (2.8)	430 (96.2) 10 (76.9)	17 (3.8) 3 (23.1)	0.015
Wound healing deficit	No Yes	391 (85.0) 69 (15.0)	376 (96.2) 64 (92.8)	15 (3.8) 5 (7.2)	0.202
Bile fistula, bile leakage, bilioma	No Yes	369 (80.2) 91 (19.8)	359 (97.3) 81 (89.0)	10 (2.7) 10 (11.0)	0.002
Liver parenchyma	Normal	163 (35.4)	156 (95.7)	7 (4.3)	0.167
	Steatosis	145 (31.5)	142 (97.9)	3 (2.1)	
	Fibrosis	43 (9.3)	42 (97.7)	1 (2.3)	
	Cirrhosis	26 (5.7)	24 (92.3)	2 (7.7)	
	Not classified	83 (18.0)	80 (96.4)	3 (3.6)	
Ascites	No Yes	355 (77.2) 105 (22.8)	349 (98.3) 91 (86.7)	6 (1.7) 14 (13.3)	<0.001

Pleural effusion	No Yes	361 (78.5) 99 (21.5)	353 (97.8) 87 (87.9)	8 (2.2) 12 (12.1)	<0.001
Revision surgery	No Yes	415 (90.2) 45 (9.8)	404 (97.3) 36 (80.0)	11 (2.7) 9 (20.0)	<0.001
Organic failure	No Yes	317 (68.9) 143 (31.1)	314 (99.0) 126 (88.1)	3 (1.0) 17 (11.9)	<0.001
Electrolyte disorders	No Yes	254 (55.2) 206 (44.8)	253 (99.6) 187 (90.8)	1 (0.4) 19 (9.2)	<0.001
Postoperative delirium	No Yes	411 (89.4) 49 (10.6)	394 (95.9) 46 (93.9)	17 (4.1) 3 (6.1)	0.460
Infections	No Yes	337 (73.3) 123 (26.7)	333 (98.8) 107 (87.0)	4 (1.2) 16 (13.0)	<0.001
Sepsis	No Yes	433 (94.1) 27 (5.9)	424 (97.9) 16 (59.3)	9 (2.1) 11 (40.7)	<0.001
CHA_2_DS_2_-VASc-Score ≥ 4	No Yes	376 (81.7) 84 (18.3)	365 (97.1) 75 (89.3)	11 (2.9) 9 (10.7)	0.004
New-onset PA	No Yes	435 (94.6) 25 (5.4)	422 (97.0) 18 (72.0)	13 (3.0) 7 (28.0)	<0.001

## Data Availability

The datasets used and/or analyzed during the current study are available from the corresponding author upon reasonable request.

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
