# Peer review of "Short- and Long-Term Outcomes of Patients with Postoperative Arrhythmia after Liver Surgery"

_biomedicines, 2024, doi:10.3390/biomedicines12020271_

Round 1

Reviewer 1 Report

Comments and Suggestions for Authors

it is an interesting and difficult and tedious study which has been wee carried out. the authors could not however determine succinctly whether postoperative arrhthmia was a cause or effect of postoperative liver complications and should state this clearly in the conclusion.

Author Response

it is an interesting and difficult and tedious study which has been wee carried out. the authors could not however determine succinctly whether postoperative arrhthmia was a cause or effect of postoperative liver complications and should state this clearly in the conclusion.

Thank you for your feedback and for making this point. We have added this in our conclusion part. 

Reviewer 2 Report

Comments and Suggestions for Authors

The authors report on the impact of postoperative arrhythmia complicating liver surgery on in-hospital mortality and long-term prognosis.

 Although this is a very interesting report examining the impact of PA after liver surgery, too many parameters involved in PA. There is also no mention of the need for prevention or treatment of PA. It may be the onset of PA due to severe complications, which may not be an independent factor.

 Comment on the following.

 â‘ Results:

l  May be a request to the editorial board, Figure 1 is incorrectly positioned and is next to Table 1

 l  In Tables 1 and 3.

     Please provide gender.

 Many parameters are confounded and should be simpler.

 Many liver procedures are presented, can they be reduced?

 What is the definition of major resection? 

   What is the difference between Infection and sepsis?

    Can you consolidate the parameters of coronary heart disease, myocardial      infarction, heart failure and valvular heart disease into one parameter?

     What are intraoperative complications?

Operating time, blood loss and transfusion( Discussion lines 263-265 are concerned.) should be added.

      Many of the factors of the CHA2DS2-VASc-score are confounded by the          data-collected parameters. Why was it adopted as one parameter ?

  l  Figure 2 and Table 4 show the same data; Table 4 should be deleted.

â‘¡Discussion:

l  The sentence from line 276 states that PA shows poor survival in logistic regression analysis, However, the results show that the PA only accepts a poor prognosis in the Kaplan-Meier curve.

 l  Are there any prevention or treatment assessments for PA?

 l  Line151 Spell out ‘Mdn’ ‘IQR’

 Line258 same ‘POAF’

Author Response

There is also no mention of the need for prevention or treatment of PA. It may be the onset of PA due to severe complications, which may not be an independent factor.

Thank you for this point. We totally agree that prevention and treatment of PA are crucial issues. Since this is a retrospective analysis we could not address these aspects. Currently we are planning a prospective study where prevention and treatment will be key parameters.

â‘ Results:

In Tables 1 and 3.

Please provide gender.

Gender has been added as second row in each table

Many parameters are confounded and should be simpler.

We decided to list all study parameters in Table 1 and 3 since this is an explorative study. We agree that some parameters might be confounded. We decided now to drop “Hereditary coagulation disorders”, “Obstructive airway diseases”, “Obstructive sleep apnoea syndrome” and “Thyroid disorders” to focus on the relevant parameters.

Many liver procedures are presented, can they be reduced?

Since these are procedures with a quite different extent of resection, we would not tend to build larger groups. Nevertheless, the classification as “major/minor resection” is one way of dividing the procedures into only two large groups. We hope you are satisfied with this solution.

What is the definition of major resection? 

The classification of the extent of liver resection was based on that of Reddy et al. (Reference 11; Reddy SK, Barbas AS, Turley RS, Steel JL, Tsung A, Marsh JW, Geller DA, Clary BM. A standard definition of major hepatectomy: resection of four or more liver segments. HPB (Oxford). 2011 Jul;13(7):494-502. doi: 10.1111/j.1477-2574.2011.00330.x. PMID: 21689233; PMCID: PMC3133716.): Operations in which fewer than four liver segments were resected were considered minor resections (vs. four or more segments in major resections). We added this answer in the method section. Thank you for pointing this out.

What is the difference between Infection and sepsis?

We added the following answer in the Method section:

In order to identify as many patients with sepsis as possible, first, the qSOFA score was calculated as the sum of the following (fulfilled) clinical criteria (Singer et al. 2016):

    • ) Respiratory frequency: ≥ 22/min
    • ) Systolic blood pressure: < 100 mmHg
    • ) CNS: reduced vigilance or altered mental status (GCS score)

If at least two of these criteria applied to a patient, septic organ dysfunction was suspected and the simultaneous presence of infections (documented by a doctor in the patient file) and organ failure was checked as further evidence.

Can you consolidate the parameters of coronary heart disease, myocardial      infarction, heart failure and valvular heart disease into one parameter?

“Preexisting cardiac illness” is now the suggested resulting parameter of these variables (explained in Method section)

What are intraoperative complications?

In this study intraoperative surgical complications include e.g. iatrogenic perforation of the liver capsule or accidental injury to vessels or biliary tract and/or need for transfusion

Operating time, blood loss and transfusion( Discussion lines 263-265 are concerned.) should be added.

Blood loss and transfusion are included into intraoperative surgical complications. Operating time is associated with the extend of liver resection and therefore not measured separately.

Many of the factors of the CHA2DS2-VASc-score are confounded by the          data-collected parameters. Why was it adopted as one parameter ?

In a metanalysis of Chen et al. the CHA2DS2-VASc-score was found to be an independent predictor of POAF after cardiac surgery. It is also associated with mortality and complications after noncardiac surgery (Sung et al. 2021). Consequently, since there is evidence in research, we adopted the CHA2DS2-VASc-score as one parameter.

  • Literature:
    • CHA2DS2-VASc-score for identifying patients at high risk of postoperative atrial fibrillation 2020
    • Septicemia and mortality after noncardiac surgery associated with CHA2DS2-VASc-score: a retrospective cohort study based on a real-world database 2021

l  Figure 2 and Table 4 show the same data; Table 4 should be deleted.

Table 4 was deleted and the reference in the text adjusted.

â‘¡Discussion:

l  The sentence from line 276 states that PA shows poor survival in logistic regression analysis, However, the results show that the PA only accepts a poor prognosis in the Kaplan-Meier curve.

The Cox regression model is referring to the Kaplan-Meier curve. 

l  Are there any prevention or treatment assessments for PA?

We are not aware of any established prevention methods for PA. According to some studies, the fluid substitution could be a factor. Also we suspect that balanced electrolytes might help preventing PA. However, since none of those are analyzed properly in a prospective study, possible prevention of PA remains subject to future studies. Treatment assessment for PA are established for cardiothoracic surgery (Hindricks G, Potpara T, Dagres N, Arbelo E, Bax JJ, Blomström-Lundqvist C, Boriani G, Castella M, Dan GA, Dilaveris PE, Fauchier L, Filippatos G, Kalman JM, La Meir M, Lane DA, Lebeau JP, Lettino M, Lip GYH, Pinto FJ, Thomas GN, Valgimigli M, Van Gelder IC, Van Putte BP, Watkins CL; ESC Scientific Document Group. 2020 ESC Guidelines for the diagnosis and management of atrial fibrillation developed in collaboration with the European Association for Cardio-Thoracic Surgery (EACTS): The Task Force for the diagnosis and management of atrial fibrillation of the European Society of Cardiology (ESC) Developed with the special contribution of the European Heart Rhythm Association (EHRA) of the ESC. Eur Heart J. 2021 Feb 1;42(5):373-498. doi: 10.1093/eurheartj/ehaa612.).

 l  Line151 Spell out ‘Mdn’ ‘IQR’

These parameters were spelled out.

 Line258 same ‘POAF’

This parameter was also spelled out.

Round 2

Reviewer 2 Report

Comments and Suggestions for Authors

The author responded well to the comments and corrected the paper.

I consider the acceptance to be "biomedicine".